

# Precopulatory behavior and sexual conflict in the desert locust

Yiftach Golov[1], Jan Rillich[1], Ally Harari[2] and Amir Ayali[1]

[1] School of Zoology, Tel Aviv University, Tel Aviv, Israel
[2] Department of Entomology, Volcani Center, Bet Dagan, Israel

## ABSTRACT

Studies of mating and reproductive behavior have contributed much to our understanding of various animals' ecological success. The desert locust, *Schistocerca gregaria*, is an important agricultural pest. However, knowledge of locust courtship and precopulatory behavior is surprisingly limited. Here we provide a comprehensive study of the precopulatory behavior of both sexes of the desert locust in the gregarious phase, with particular emphasis on the conflict between the sexes. Detailed HD-video monitoring of courtship and mating of 20 locust pairs, in a controlled environment, enabled both qualitative and quantitative descriptions of the behavior. A comprehensive list of behavioral elements was used to generate an eight-step ethogram, from first encounter between the sexes to actual copulation. Further analyses included the probability of each element occurring, and a kinematic diagram based on a transitional matrix. Eleven novel behavioral elements are described in this study, and two potential points of conflict between the sexes are identified. Locust sexual interaction was characterized by the dominance of the males during the pre-mounting stage, and an overall stereotypic male courtship behavior. In contrast, females displayed no clear courtship-related behavior and an overall less organized behavioral sequence. Central elements in the sexual behavior of the females were low-amplitude hind-leg vibration, as well as rejecting males by jumping and kicking. Intricate reciprocal interactions between the sexes were evident mostly at the mounting stage. The reported findings contribute important insights to our knowledge of locust mating and reproductive behavior, and may assist in confronting this devastating agricultural pest.

## INTRODUCTION

The desert locust, *Schistocerca gregaria* (Forskål) (Orthoptera, Acrididae) is one of the most serious agricultural pests. During outbreaks, swarms may consist of millions of individuals (40–80 million locusts per km$^2$; e.g., *Singh & Singh, 1977*; *Steedman, 1988*; *EL-Bashir et al., 1993*), and the damage to crops can be enormous, as the locusts are able to consume hundreds of tons of vegetation per day (*Shaluf, 2007*). Moreover, according to some estimates, 1/10 of the global human population is affected by this pest (*Latchininsky et al., 2011*).

Locusts have served as important models in the study of various aspects of insect physiology and behavior (e.g., *Burrows, 1996*; *Ayali & Yerushalmi, 2010*; *Ayali & Lange, 2010*; *Ariel & Ayali, 2015*). However, our knowledge of locust courtship and precopulatory

Corresponding author
Amir Ayali, ayali@post.tau.ac.il

behavior is surprisingly limited. Understanding the mating and reproductive behaviors of a species has a fundamental role in the understanding of its ecological adaptation (*Kirkendall, 1983*; *Thornhill & Alcock, 1983*). Specifically, thorough comprehension of the interactions between the sexes may provide new insights for the development of alternative methods for coping with agricultural pests (*Boake, Shelly & Kaneshiro, 1996*; *Suckling, 2000*). This should be achieved by a combination of qualitative descriptions and quantitative analyses—the two complementary components of an ethological study (*Kasuya, 1983*).

A comprehensive study of animal behavior should start with a list of behavioral elements (or 'units'), followed by their chronological appearance, in order to construct a species-specific **ethogram**. The quantification of behavioral elements needs to be based not only on the appearance of these elements, but also on their frequency, their sequence and the probability of transition. Such an approach can identify the typical elements and key transitions during the behavioral ritual (e.g., *Klein & De Araújo, 2010*). The quantification can be aided by using a Markovian chain, also known as a transition matrix (*Castrovillo & Cardé, 1980*; *Haynes & Birch, 1984*). The knowledge gained may contribute not only to deciphering evolutionary relationships between taxa, as in host-parasite interactions, but also to the understanding of mate recognition and sexual conflict (*Paranjape, 1985*; *Curkovic, Brunner & Landolt, 2006*; *Cozzie & Irby, 2010*; *Gaertner et al., 2015*), and specifically so in agricultural pests (*Walgenbach & Burkholder, 1987*; *Rojas et al., 1990*; *Wang & Millar, 2000*; *Zahn et al., 2008*).

Some aspects of the sexual behavior of *S. gregaria* have been previously addressed (*Uvarov, 1928*; *Husain & Mathur, 1946*; *Laub-Drost, 1959*; *Laub-Drost, 1960* cited in *Uvarov, 1966*; *Uvarov, 1977*; *Popov, 1958*; *Loher, 1959*; *Loher, 1961*; *Pener, 1965*; *Pener, 1967a*; *Pener, 1967b*; *Norris & Richards, 1964*; *Odhiambo, 1966*; *Roffey & Popov, 1968*; *Strong & Amerasinghe, 1977*; *Uvarov, 1977*; *Amerasinghe, 1978a*; *Pener & Lazarovici, 1979*; *Inayatullah, El Bashir & Hassanali, 1994*; *Njagi & Torto, 2002*), but much of the required knowledge is still lacking. The published descriptions and quantifications of the sexual behavior of both sexes are either limited (*Strong & Amerasinghe, 1977*; *Inayatullah, El Bashir & Hassanali, 1994*), too general, or focus predominantly on the male (e.g., *Pener, 1967b*; *Amerasinghe, 1978b*). In addition, previous studies suffer from inconsistencies (e.g., different names for similar behavioral elements). Finally, little effort has been dedicated to the study of sexual conflict in this insect. The desert locust displays a clear sexual dimorphism in the gregarious phase, with fully mature males being bright yellow and females being beige-brown to yellowish (*Chauvin, 1941* cited in *Pener & Simpson, 2009*; *Norris, 1954*; *Pener, 1967b*). As is the case for many other acridids, little is known regarding the means of sexual recognition in the desert locust (*Whitman, 1990*). It is postulated, however, that visual and chemical signals play an important role (*Obeng-Ofori, Torto & Hassanali, 1993*; *Obeng-Ofori et al., 1994*; *Francke & Schmidt, 1994*; *Inayatullah, El Bashir & Hassanali, 1994*; *Ould Ely et al., 2006*; *Seidelmann & Warnstorff, 2001*). Courtship and mating behaviors can be roughly divided into two sequential stages: pre-copulatory and post-copulatory (with copulation defined as the time when sperm is transferred). The pre-copulatory stage can be further divided into two further sub-stages: pre-mounting, comprising all the behavioral elements leading to a successful mounting attempt; and

mounting, culminating in successful copulation. Locust courtship is considered simple and primitive (*Loher, 1959*; *Uvarov, 1966*; *Uvarov, 1977*; *Oberlin, 1974* cited in *Strong & Amerasinghe, 1977*). As in many grasshoppers, males of *S. gregaria* have been reported to be the dominant gender during the sexual-interactions (*Norris & Richards, 1964*; *Pener, 1965*; *Pener, 1967b*; *Strong & Amerasinghe, 1977*; *Amerasinghe, 1978a*; *Inayatullah, El Bashir & Hassanali, 1994*). Briefly, the male's sexual intention is initially demonstrated through his orientation towards the female, followed by a stealthy slow approach and a surprise attempt to mount her. Once mounting, the male grasps the female using his front and mid-legs. Copulation is achieved when the male moves his abdomen along the side of the female and connection between the genitalia is established. In contrast to the males, gregarious females have been considered to demonstrate no clear courtship behavior (*Norris & Richards, 1964*; *Pener, 1965*; *Pener, 1967b*; *Strong & Amerasinghe, 1977*; *Amerasinghe, 1978a*; *Inayatullah, El Bashir & Hassanali, 1994*). Nonetheless, the rejection of courting males has been reported, including the female's jumping (before and during mounting), kicking, and lateral movements of her abdomen in the attempt to prevent copulation (*Loher, 1959*; *Strong & Amerasinghe, 1977*; *Uvarov, 1977*). Hind leg vibration and wing stridulation have been reported to be displayed during the pre-copulatory behavior (*Morse, 1896*; *Norris, 1954*; *Laub-Drost, 1959* cited in *Uvarov, 1977*; *Loher, 1959*; *Loher, 1961*; *Otte, 1970*; *Uvarov, 1966*; *Uvarov, 1977*), as in other acridid grasshoppers (*Haskell, 1957*; *Haskell, 1958*; *Otte, 1970*). Unlike wing stridulation (displayed by both sexes), the vibration of the hind legs is soundless and much more common in the female (*Loher, 1959*). The role of both behavioral elements in the sexual interaction has remained uncertain (*Loher, 1959*; *Uvarov, 1966*; *Otte, 1970*).

The major goals of this work were to generate an ethogram, comprising and accompanied by both qualitative and quantitative tools for studying the sexual behavior of the two sexes of the desert locust during the pre-copulatory stage. This included generating a detailed list of all related behavioral elements, and consolidating the relevant terminology (i.e., 'nomenclatura'). The generated ethogram includes all the behavioral elements, their occurrences, and their sequence during the sexual interaction. This enabled an elaborate description of the conflict between the sexes in gregarious locusts. We are currently employing the tools developed herein in a comprehensive investigation of the two density-dependent locust phases.

## MATERIAL & METHODS

### Animals

Desert locusts, *Schistocerca gregaria* (Forskål) (Orthoptera, Acrididae), from our colony at Tel Aviv University (*Ayali, Zilberstein & Cohen, 2002*) were reared for many consecutive generations under crowded conditions (i.e., approaching the gregarious phase), 100–160 individuals in 60 aluminum cages. All cages were located in a dedicated room under a constant temperature (29–31 °C) and light cycle of 12: 12 D: L. Supplementary radiant heat was supplied during day-time by incandescent 25 W electric bulbs (full visible spectrum, yellow and red dominant), resulting in a day temperature of c. 37 °C. Locusts were

provided daily with fresh wheat and dry oats, and plastic caps (300cc) filled with moist sand for oviposition.

All locust individuals in the experiments were adult virgin males and females. Virgin adults were obtained by marking newly-emerged adults with non-poisonous acrylic paint within 24 h following ecdysis. Males and females were separated into single sex "cohort cages" every three days. Thus, in each cohort cage the maximum age range of the individual locusts was less than 72 h. The cages were maintained under the same rearing conditions as above. For the observations we used 12-14-days-old males, when their yellowish coloration had reached stage V (see *Norris, 1954*; *Loher, 1961*). This stage is known to coincide with sexual maturity. Females were 18–20 day-old, sexually mature, based on our preliminary work and other previous reports (*Hamilton, 1955*; *Injeyan & Tobe, 1981*; *Mahamat et al., 1993*; *Wybrandt & Andersen, 2001*; *Ould Ely et al., 2006*; *Nishide & Tanaka, 2012*). Only fully intact insects participated in the observations.

## Experimental design

Experiments were carried out in an isolated room, with temperature and light conditions similar to that in the rearing room. A plastic observation cell (14 × 13 × 24 cm) was initially divided by an opaque plastic partition into two compartments, to separately host the male and the female. The sensitivity of *S. gregaria* color vision is mainly in the very short wavelengths of both UV and blue (320 and 450 nm), and to a lesser extent, also in the green range (light 530 nm) (*Eggers & Gewecke, 1993*; *Schmeling et al., 2014*). Hence, until initiation of the experiment the cell was illuminated by a red light (to reduce the insects' stress). Five minute after placing each locust (one male and one female) into its own compartment, the experiment was initiated by carefully removing the partition between the compartments, and replacing the red light with two regular 25 W light bulbs. Two identical observation cells, separated only by a dense plastic mesh (not sealed), were used simultaneously (each housing one pair of locusts), generating crowd-like conditions by allowing the flow of auditory, olfactory and visual cues. Experiments lasted 3 h, or until copulation had occurred, if earlier, and were recorded by a SONY HDR-PJ820E video camera.

Two rounds of experiments were carried out daily: at 08:00 AM and 15:00 PM. Out of an overall 31 monitored experiments, 20 ended in copulation within the defined time of 3 h, and were used in the analyses.

In order to further verify the significance of the females' active rejection behaviors (jumping and kicking) and their possible roles in female choice, a separate series of experiments were carried out. Here we examined male mating success when facing "handicapped" females. The rejection attempts of these females were constrained by means of a small rubber band confining the hind legs' femur and tibia in a folded position, and thus, preventing the female from either jumping or kicking. The number of male mounting attempts and successful mounts were compared between pairs of males and constrained females ($N = 10$) and males and unrestrained females ($N = 20$).

## Data analyses

The recorded videos of the behavior of each pair were reviewed and analyzed using J-watcher software (version 0.9 for Windows).

Behavioral elements were identified in order to describe the locusts' pre-copulatory behavior. These included both repetitive (lengthy, e.g., the vibration of the hind leg femur) and discrete (momentary, e.g., jumping) behaviors. The two behavioral types were counted, with a 'count' relating to the duration of a behavior from initiation until termination. Behavioral measurements were taken only if the male and female were at a distance of less than 10 cm (i.e., an 'encounter'). For both pre-mounting and mounting behaviors, the following parameters were measured and compared for both sexes: (1) in order to obtain the pattern or chronological sequence of the behavioral repertoire, the relative time to initiation of each behavior was noted (relative to the total time of the relevant stage, either pre-mounting or mounting). (2) The probability of a specific behavior occurring (PO = 1 if the behavior occurred at least once, and 0 otherwise). (3) The frequency of occurrences of a specific behavioral element.

A kinematic diagram was constructed, based on a first-order Markov model, for all the transitions between pairs of behavioral elements (i.e., preceding–following elements) that are mutually exclusive (*Baker & Cardé, 1979*). All the behavioral elements in this analysis were considered nodes and used to construct a transitional matrix. The transition probability (TP, also known as 'conditional probability'; *Wood, Ringo & Johnson, 1980*) was first calculated based on all possible transitions between a pair of nodes in the matrix, for each experiment (see also *Brown, 1974*; *Leonard & Ringo, 1978*; *Markow & Hanson, 1981*). Next, the average of each transition was calculated among all 20 pairs for each sex (following the method described by *Charlton & Cardé, 1990*). Overall, the behavioral transitional matrix comprised 25 elements for the male, and 18 for the female. Self-transitions were scored as structural zeroes (*Baker & Cardé, 1979*), and impossible transitions were left blank (*Haynes & Birch, 1984*). Those behavioral elements that were not mutually exclusive with any of the other elements ('antennal movement', 'palp vibration', 'genital-opening', 'abdominal wagging') were excluded from this analysis. Transitional probabilities (i.e., TP) ≤10% are not presented. The total sum of transitions from a given element may exceed 100% in cases where an element was followed by at least two elements which were not mutually exclusive.

Most of the statistical output and data analysis were conducted in GraphPad Prism version 6.04 for Windows, JMP®, Version 12.0.1 SAS Institute, and some in Matlab (Math-Works, Inc., Natick, MA, USA) and Canvas draw 2.0 (Deneba Systems, Miami, FL, USA).

## RESULTS

### The sexual behavior of the desert locust

As noted above, the behavioral elements that lead to copulation (i.e., those that can be identified during the pre-copulatory phase), can be divided into two stages: Table 1 lists all the elements comprising the pre-mounting stage, and Table 2 lists all the elements comprising the mounting stage (ending in copulation). Within the behavioral repertoire

**Table 1** The behavioral repertoire during the pre-mounting stage.

| Behav. element | Description | Sex | | Citation |
|---|---|---|---|---|
| | | m | f | |
| Grooming | Eyes, antennas, mouth part, mid and front legs, wings and abdominal grooming | + | + | *O'shea (1970)*, *Rowell (1971)*, *Uvarov (1977)* and *Berkowitz & Laurent (1996)* |
| Antennal movement | Movement of the antennas | + | + | *Loher (1959)* and *Wallace (1959)* |
| Palp vibration | Vibration movements of both labial and maxillary palps | + | + | *Uvarov (1977)* |
| Searching | Combination of walking, scanning, antennal movement and palps vibration | + | + | Based on Inayatullah et al. (1996) |
| Orienting | Anterior side is directed towards the female | + | − | *Otte (1970)*, *Amerasinghe (1978a)* and *Amerasinghe (1978b)* |
| High hind leg (HL) femoral vibration | Femur is lifted to a perpendicularly position relatively to the ground (∼90°) while vibrating the hind legs. | + | + | *Loher (1959)* |
| Low hind leg (HL) femoral vibration | Femur is lifted to a perpendicularly position relatively to the ground (<90°) while vibrating the hind legs. | + | + | *Loher (1959)* |
| Wing flapping | Flapping movements of both wing pairs | + | + | *Weis-Fogh (1956a)*, *Weis-Fogh (1956b)* cited in *Uvarov (1966)*; *Uvarov (1977)* |
| Long stridulation | Rapid vibration of the wing-pairs producing whizzing noise | + | − | *Loher (1959)* |
| Short stridulation | Wings beating against | + | − | *Loher (1959)* |
| Abdomen wagging | Wagging (mostly lateral) movements of the abdomen | + | + | [a] |
| Genital opening | Rhythmic opening of the genital-opening | + | + | [a] |
| Initiating physical contact | Locusts touching each other | + | + | *Popov (1958)* |
| *Mutual antennal contact* | Locust antennas touching each other | + | + | [a] |
| Slow repeating hind leg elevation | Slow elevation of the hind legs | + | − | [a] |
| Approaching | Walking clearly directed towards other sex | + | + | *Popov (1958)* and *Loher (1959)* |
| Walking away from the male | Distancing from the male by walking | − | + | *Popov (1958)* and *Loher (1959)* |
| Jumping away from the male | Distancing from the male by jumping | − | + | *Popov (1958)* and *Loher (1959)* |
| Mounting by Climbing | Attempting to mount the female by climbing | + | − | [a] |
| Mounting attempt by Jumping | Attempting to mount the female by jumping | + | − | *Uvarov (1928)* and *Husain & Mathur (1946)* |

**Notes.**
Behaviors which are shared and mutually exhibited by both sexes are presented in italic, bold font.
[a]Represents elements which are described for the first time.

**Table 2  The behavioral repertoire during the mounting stage.**

| Behav. element | Description | Sex | | Citation |
|---|---|---|---|---|
| | | **m** | **f** | |
| *Mounting* | The male mounting the female/ female is being mounted by the male | + | + | *Uvarov (1928)*; *Husain & Mathur (1946)* |
| Avoidance | Elevation of the two hind legs femur close together in the air | + | − | [a] |
| Blocking | Lifting both Hind legs in the air perpendicularly to the ground, femur and tibia folded | + | − | [a] |
| Jumping | Jumping while carrying the male | − | + | *Loher (1959)* |
| Kicking the male | Directed kicking towards the male | − | + | *Loher (1959)* |
| Abdominal bending | Abdominal tips and genital are bended sideways | − | + | *Strong & Amerasinghe (1977)* |
| Abdomen grounding | Abdomen is pressed to the ground | − | + | [a] |
| *Male's dislodgement* | Male is dislodged from the female's back | + | + | *Popov (1958)* and *Uvarov (1977)* |
| Lifting attempt | Males attempt to lift the female by pushing against the ground and straightening of the hind legs | + | − | [a] |
| Antennal movement | Movement of the antennas | + | + | *Uvarov (1977)* |
| Palp vibration | Vibration of both labial and maxillary palps | + | + | *Loher (1959)* |
| Hind legs vibration | Femur is being positioned perpendicularly to the ground while tibia is flexed 90° to the femur. | + | − | *Loher (1959)* |
| Short Stridulation | Wings beating against each other resulting in short and sharp sounds | + | − | *Loher (1959)* |
| Long stridulation | Rapid vibration of the wings while in resting position | + | − | *Loher (1959)* |
| Lateral hind leg vibration | Lateral vibration of the hind legs femur | − | + | *Loher (1959)* |
| Hind leg grounding | Pushing the hind legs against the ground. | + | − | [a] |
| Genital opening | Rhythmic opening–closing of the genital opening | + | + | [a] |
| Copulation attempt | Male's abdomen is placed to the side of the female's ones in order to reach her genitalia | + | − | *Loher (1959)* |
| *Copulation* | Copulation defined as the time when sperm is transferred | | | *Pener (1967a)* and *Pener (1967b)* |

**Notes.**
Behaviors which are shared and mutually exhibited by both sexes are presented in italic, bold font.
[a]Represents elements which are described for the first time.

listed in Tables 1 and 2, several elements have been described previously. However, those descriptions tend to be episodic, with different authors providing different descriptions for the same behavior, or referring to the same behavior by different names, etc. Eleven elements are novel, and are described here for the first time.

The probability of each element being demonstrated varies greatly. Figure 1 denotes the probability of a behavioral element occurring (PO), separately for males and females for the pre-mounting and mounting stages. The behavioral elements appear in a consecutive order and are grouped following a further subdivision: S1–S7, from initiation (S1) to copulation attempt (S7), culminating in S8, copulation. Fig. 1 presents the elements that involve all body parts (denoted by different colors), including legs, wings, palps and antennae, and abdomen. Some of the sub-stages are characterized by a consistently high PO (e.g., S1 during the pre-mounting; Fig. 1) while that of others varies greatly. Moreover, the PO of the elements demonstrated by the male or the female within the same sub-stage differs

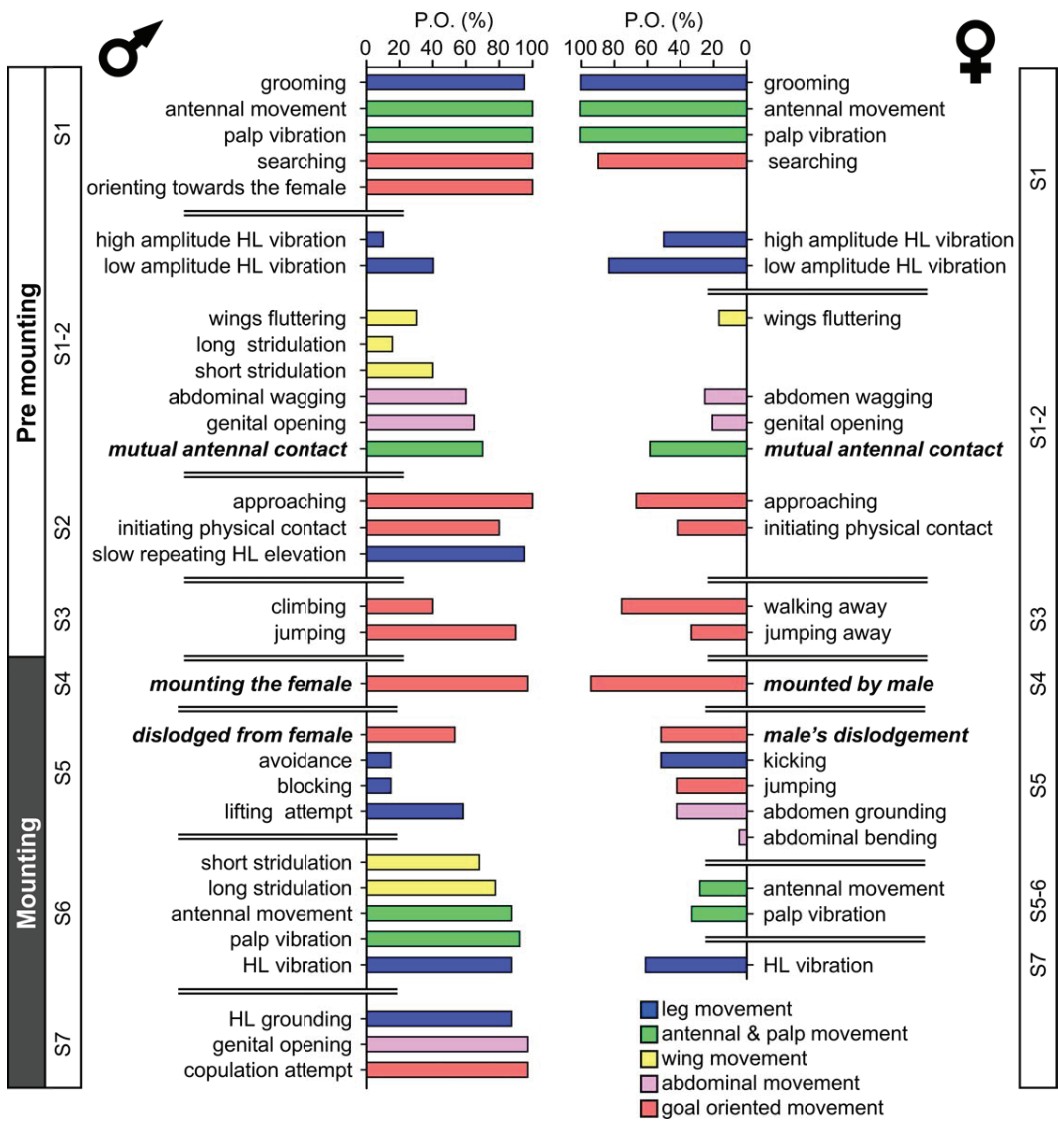

**Figure 1 The precopulatory behavioral repertoire of the male and female desert locust.** The premounting and mounting behavioral elements are listed from step 1 to 7 (S1–S7) and color coded according to relevant body part. The mean probability of an element occurring (PO, %) is shown. Behavioral elements that are shared and mutually exhibited by both sexes are presented in italic bold font.

(e.g., compare S1-2 or S5-7 in Fig. 1). Generally speaking, a high PO reflects the importance of a behavioral element within the overall sequence. However, there may be low PO elements that nonetheless have a crucial functional significance: e.g., those instrumental in inter- and probably also intra- sexual communication (e.g., hind leg vibration, wing flutter and stridulation). Illustrations of the different behavioral elements are provided in Fig. 2.

An ethogram was constructed (Fig. 3) in order to better characterize the behavioral sequence comprising the pre-copulatory behavior. The ethogram provides the pre-mounting and mounting stages (consistent with Fig. 1), presenting them as an ordered,

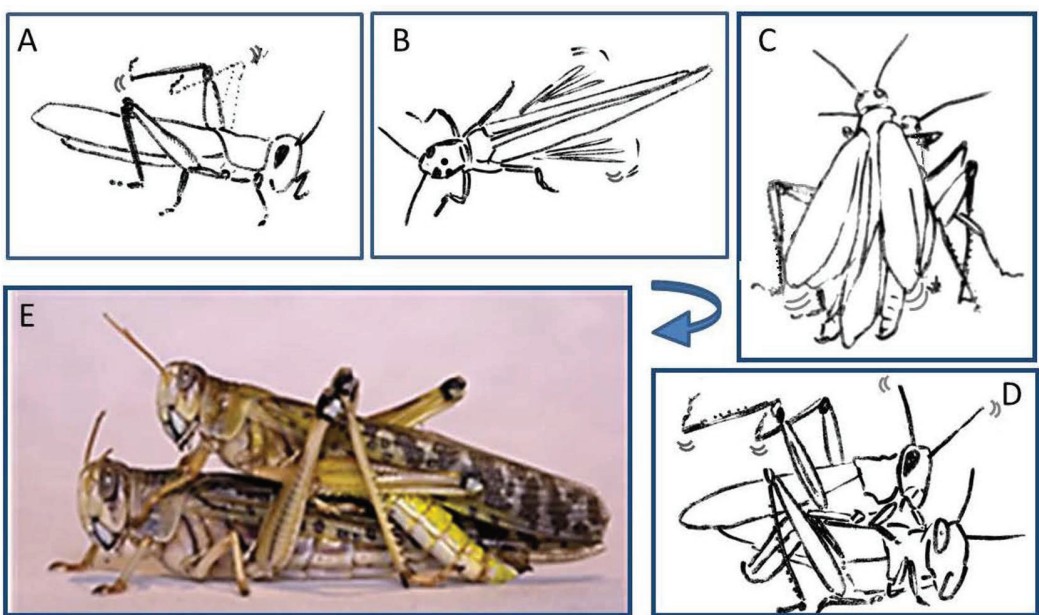

**Figure 2 Representative behavioral elements observed during the pre-mounting (A–B) and mounting (C–E) stages** (A) Male slow repeated hind leg elevation. (B) Female hind leg low and high-amplitude vibration. (C) Male short and long wing stridulation. (D) Male hind leg vibration and copulation attempt. (E) Successful copulation. The animations in (A–D) were drawn from images taken from video sequences.

hierarchical flow chart. This representation also allowed us to include and emphasize junctions or decision points (denoted by the traffic lights in Fig. 3). These junctions represent the culmination of the conflict between the sexes, e.g., a point at which the female was successful in preventing a mounting attempt by jumping away, or a point at which the male was thrown off the female's back. Illustrations of behavioral elements of antagonistic nature can be seen in Fig. 4.

Further information regarding the flow of the behavioral elements and the overall sequence of the behavior can be obtained by also including, beyond the ordered description of the elements, the probability of a transition from one element to the other. This approach regards the behavioral sequence as the Markov process or Markov chain, in which the appearance of each behavioral element affects or predicts the probability of the appearance of another. Figures 5 and 6 use a similar color code as that presented in Fig. 1 to indicate the different behavioral elements constituting the sub-stages (S1–8), presented in Figs. 1 and 3. These kinematic diagrams denote a weighted directed network composed of the above introduced different behavioral elements presented by males (Fig. 5) and females (Fig. 6), where the weights are the transition probabilities (TP). As can be seen, this method of presentation clearly discriminates between behavioral elements constituting the relatively consistent or major trunk (depicted 0–8 in Fig. 5, and 0–5 in Fig. 6), as well as the various possible detours or diversifications from it. It also serves to highlight several sex-specific characteristics, as discussed below.

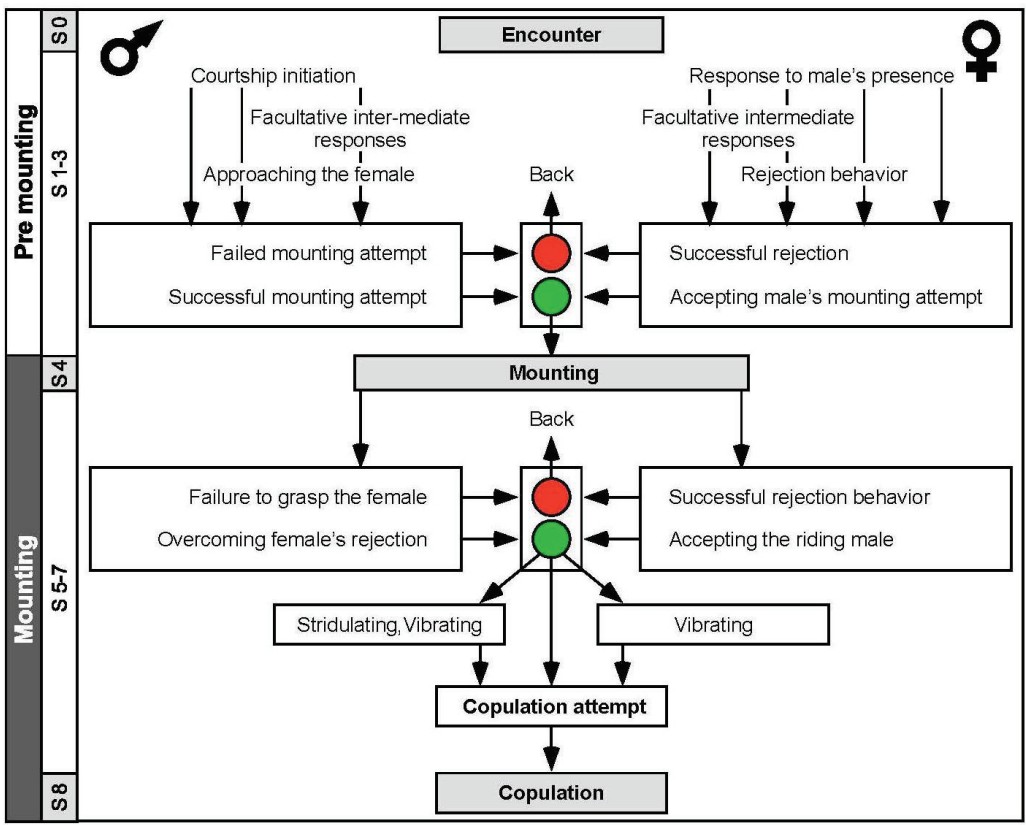

**Figure 3** **An ethogram depicting the desert locusts pre-copulatory interactions leading to copulation.** The male behavior is on the left, and that of the female on the right. S1–S8 indicate the chronological step number during the pre-mounting and mounting stages. Traffic lights denote points at which female choice takes place (steps 3 and 6); red is associated with rejection of the male. Green is associated with the female tolerating the male.

In the following we provide further details of certain male- and female-specific behavioral elements, as well as further insights into the conflict between the sexes.

## Sex-specific sexual behaviors and conflict between the sexes

The strategy employed by males during pre-mounting can be described as stalking, pursuit and attack. Overall courtship in our experiments was somewhat limited. Upon identifying the female, the male commonly demonstrated 'high-stepping walking' behavior, carrying his body high above the ground. In some cases, this was intensified prior to jumping in an attempt to mount the female to an extent that his front legs were raised in the air. Increased self-grooming was shown by all males (PO = 100%); males groomed the antennae, the compound eyes, the front or mid pairs of legs and the posterior part of the abdomen. Several behavioral elements, commonly shown during pre-mounting, are described here for the first time. These comprise: lateral wagging movements of the abdomen ('abdominal wagging'), repeated extension movements of the subgenital-plate and the epiproct ('genital–opening'), and repeated slow elevation of the hind legs. The

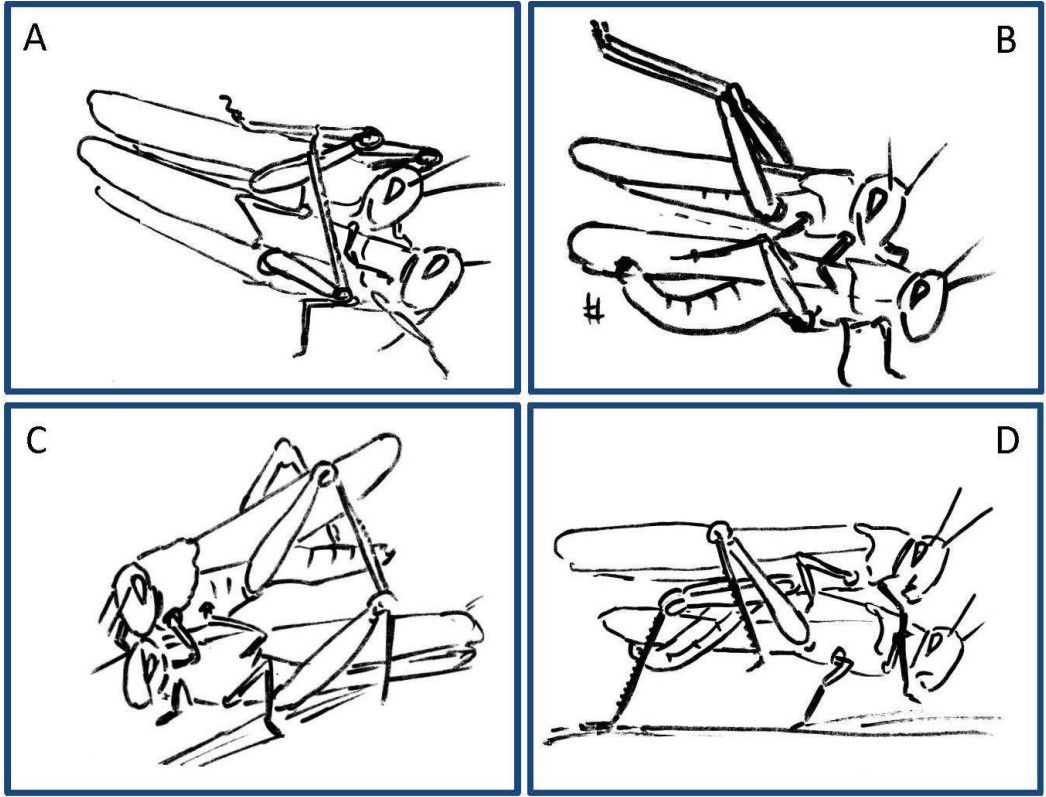

**Figure 4** Examples of female rejection behaviors and male responses during the second point of mate choice (second traffic light in the ethogram in **Fig. 3**). (A) Male attempts to block the female's kicking using his hind legs. (B) Female displaying lateral abdomen bending behavior while also kicking, and male responding to kicking by avoidance behavior. (C) Female pressing her abdomen to the ground to avoid mating (i.e., 'abdominal grounding'). (D) Male managing to mate with the female by pushing with his hind legs and lifting her.

latter was performed by most males (PO = 95 ± 5%) just after (TP = 14%) or before (TP = 11%) approaching the female.

Once successful in mounting the female's back, mostly via jumping, the majority of males were quick to cling to the lateral sides of her pronotum socket (or its edges; see Figs. 2D and 2E) in order to adjust their grip. Stridulation and hind leg vibrations were more frequent during mounting than pre-mounting, although the cumulative time of mounting (1.65 ± 0.41 min) was much shorter than the pre-mounting (64.23 ± 10.93 min).

The females' overall sequence of behavioral elements was much less stereotypic compared to that of the males (as also evident from Figs. 1, 5 and 6). In spite of the dominant part played by males, the first indication of encounter was usually demonstrated by females (17 out of 20 pairs). Hind leg vibration was a characteristic element of females pre-copulatory behavior, as demonstrated by the high values of both PO and TP (Figs. 1 and 6). However the most dominant behavioral feature was the female rejection of the males (Fig. 4).

During the pre-mounting stage, female rejection was displayed by either jumping or walking away from the male. 'Walking away' (PO 75%) was commonly followed by the

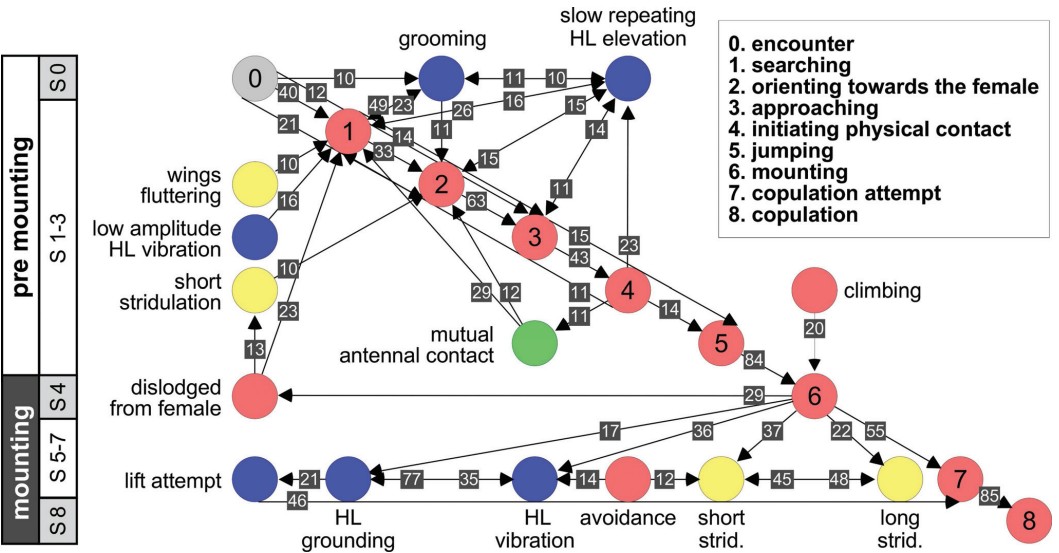

**Figure 5** A kinematic diagram depicting the pre-copulatory behavior of male locusts ($N = 20$); arrows represent transitions between behavioral elements. The numbers on a gray background denote the mean transitional probability (TP, %) between each pair of behavioral elements. Two way transitions are depicted by double-headed arrows (numbers relate to the closer arrow head). The color of the circles representing the different behavioral elements corresponds to the color index used in Fig. 1. The different steps in the pre-mounting and mounting stages are noted.

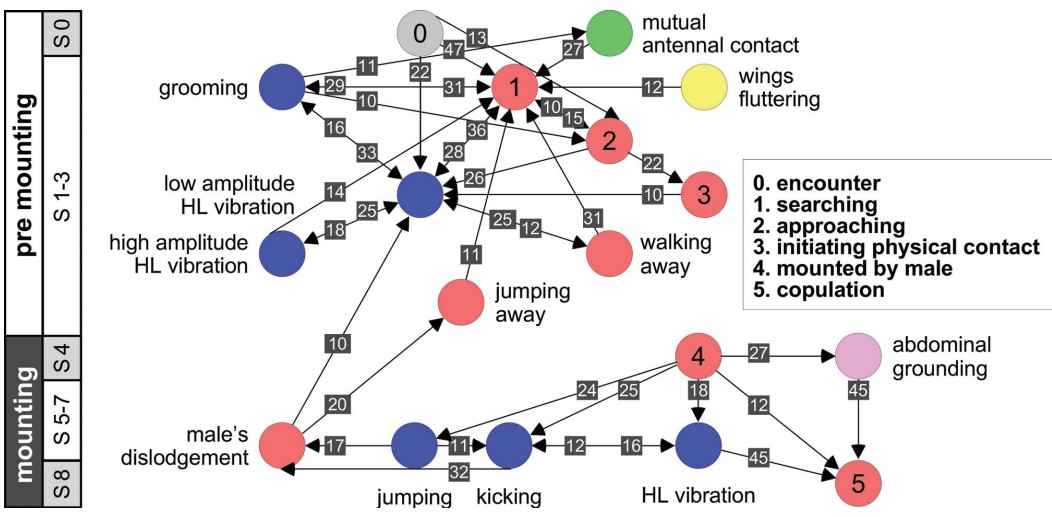

**Figure 6** A kinematic diagram depicting the sexual behavior of female locusts ($N = 20$); details as in Fig. 5.

lower amplitude hind leg vibration (TP = 25%). The most common rejection element during the mounting stage was kicking (PO = 55%). Both kicking and jumping often caused the mounted male to lose his grip and dislodge from the female's back. In fact, more than half of the mated males were dislodged from the female (PO = 55%). Females
also exhibited "passive rejection" elements, including pressing the abdomen against the ground and thus preventing the male from inserting his abdomen below hers ('abdominal grounding,' described here for the first time; Fig. 4C) and less frequently lateral abdominal bending (Fig. 4B; PO = 5%), which was very efficient in preventing copulation.

Male behavioral elements that were intended to avoid or overcome female rejection are also described here for the first time. These comprised: attempting to block the female's kicks with the male's own hind legs (Fig. 4A), and elevation of the hind legs with the tibia extended, while keeping the legs close together, in order to minimize the area exposed to the female's kicking (avoidance; Fig. 4B). Naturally, males occupied with these defensive behaviors could not progress toward copulation. An intriguing newly described element is that of the male's attempt to overcome the female abdominal grounding behavior by pushing with his hind legs and lifting her up (Fig. 4D). This reciprocal interaction is shown in a specific path of transitions in Fig. 5- 'mounting' → 'grounding of the hind legs' → 'lifting attempt' → 'copulation attempt'.

In order to further explore the selected "female choice" stages (S3 and 5; Fig. 3) and verify the significance of female active rejection behaviors and their effects on the males' mating success, we prevented females from jumping and kicking by means of a small rubber band over their folded hind legs. This manipulation indeed resulted in no active rejection by the constrained females. Consequently, the number of male mounting attempts on these females was significantly lower than that in the control group (med = 1<2, $U = 40.50$, $N_1 = 10$, $N_2 = 20$; $p < 0.01$), and 100% of the pairs comprising a constrained female and a normal male ended in copulation. The males that mounted constrained females displayed none of the documented defense behaviors (see above).

## DISCUSSION

The sexual behavior of the desert locust *S. gregaria* has been previously addressed in various studies (e.g., *Uvarov, 1928*; *Uvarov, 1977*; *Norris & Richards, 1964*; *Pener, 1965*; *Odhiambo, 1966*; *Roffey & Popov, 1968*; *Amerasinghe, 1978a*; *Pener & Lazarovici, 1979*; *Njagi & Torto, 2002*), and was mostly described as primitive and reduced (*Popov, 1958*; *Loher, 1959*; *Strong & Amerasinghe, 1977*; *Pener & Shalom, 1987*; *Inayatullah, El Bashir & Hassanali, 1994*). None of those studies, however, were dedicated to a synchronized, comprehensive investigation of the behavior of the two sexes and their sexual interactions. *Loher (1959)* for example, although devoting much effort to describing locust sexual behavior, did not include any quantitative measures of the different behavioral elements. In a first attempt to quantify the pre-copulatory behavior of the male, *Pener (1967a)* and *Pener (1967b)* employed a measure of "average percentage of time spent on sexual behavior", but with sexual behavior comprising only copulation, sexual attack, or mounting another locust. In a later report, recording the time spent in sexual behavior, *Wajc & Pener (1969)* noted the great need for elaborate quantitative methods in the study of the sexual behavior of *S. gregaria*. While other quantification efforts (e.g., *Inayatullah, El Bashir & Hassanali, 1994*) presented some accounts of behavioral elements, they provided only limited descriptions of the pre-copulatory behavior in a rather anecdotal manner, and similar to previous work focused mostly on males.
In the current study we provide in-depth data on the pre-copulatory behavior of the desert locust in the gregarious phase, comprising both qualitative descriptions and quantitative measures. A detailed list of behavioral elements is presented, incorporating eleven elements that are described here for the first time. An ethogram of the sexual behavior of both sexes, from first encounter until copulation, has enabled us to describe the dynamics of the behavior, including the probability of each element being demonstrated and the transitions between elements. Overall eight distinct steps were identified as comprising the two pre-copulatory stages: pre-mounting (S1-3) and mounting (S4-8). Most importantly, two points of conflict between the sexes were recognized and investigated in depth.

## Male sexual behavior

A major characteristic of locust sexual behavior is that of the males' dominant role in the courtship ritual (*Norris & Richards, 1964*; *Strong & Amerasinghe, 1977*; *Inayatullah, El Bashir & Hassanali, 1994*). Our findings well demonstrate that the overall initiative is always that of the male. Upon encountering a female, the males displayed a combination of self-grooming, palp vibration and antennal movements (see also *Loher, 1959*). The latter is a known characteristic of male sexual behavior in the family Acrididae (*Pickford & Gillott, 1972*; *Otte, 1970*; *Riede, 1987*). Onset of the rather limited courtship behavior can be recognized initially by the display of "orienting", in which the male points his antennae towards the female. This behavior is common in the subfamily Catantopinae (*Otte, 1970*).

Another important feature of the male pre-copulatory behavior is its relative consistency, as suggested by *Loher (1959)* for the courtship behavior of male grasshopper in all Catantopinae species. This stereotypical nature is evident in the present work from the high values of both the PO and TP quantitative measurments. Orientation was followed by a slow, stealthy approach and a sudden jump in the male's attempt to mount the female. Upon mounting, the male then displayed various stridulation and vibration behavioral elements, culminating in copulation attempts and copulation.

Overall, male sexual behavior varied more during the pre-mounting than during the mounting sub-stages. This was expressed in both the larger repertoire of elements and the higher variability of their occurrence (PO). A major behavioral element during pre-mounting was that of the slow elevation of the hind legs (described previously in males of *Aulocara elliottii*; *Bromenshenk & Anderson, 1981*). We suggest that this element reflects the internal state of the male, i.e., sexual arousal and readiness to mate (prior to mounting attempts). Limited courtship during pre-mounting was previously attributed to both *S. gregaria* (*Popov, 1958*; *Strong & Amerasinghe, 1977*) and *Locusta migratoria* (*Oberlin, 1974* cited in *Strong & Amerasinghe, 1977*). *Oberlin (1974)* suggested that this is a result of the high inter-male competition found under the crowded conditions of a locust swarm.

Stridulation (short- previously referred to as "short burst", "sharp sounds" or "assault-sounds"; and long- previously referred to as "long sounds", "long burst", or "whizzing noises"; *Loher, 1959*; *Uvarov, 1977*) and the hind leg (silent) vibration elements (referred to as "cycling of the hind legs"'; *Strong & Amerasinghe, 1977*) are known as major characteristics of male sexual pre-copulatory behavior and have been reported to

feature during both pre-mounting and mounting (see also *Norris, 1954*; Laub-Drost cited in *Uvarov, 1977*; *Otte, 1970*). While their role is still not fully resolved, in our current observations they were more frequent during mounting (as also mentioned by *Loher (1959)*). Overall, in addition to its relatively shorter duration, the mounting stage seems to be the more conserved stage in the locust's reproductive behavior.

Another intriguing behavioral element during pre-mounting is that of wing-fluttering. This was previously reported for both sexes of the desert locust during sexual interaction ("stationary wings-fluttering" in *Loher, 1959*; *Uvarov, 1966*; *Uvarov, 1977*; *Njagi & Torto, 2002*). In other acridids wing fluttering was suggested to have a role in mediating release of male volatile substances in relation to mate finding (*Uvarov, 1966*). However, the role of wing fluttering in relation to sexual behavior in the desert locust has not yet been resolved.

## Female sexual behavior

Female desert locusts demonstrated no clear courtship behavior, and were less dominant than males during the sexual interaction (also reported by *Norris & Richards, 1964*; *Strong & Amerasinghe, 1977*; *Inayatullah, El Bashir & Hassanali, 1994*). The sexual behavior of the females was also less stereotypic. Upon encountering a male, female behavior comprised palp vibration, antennal movement, searching, and self-grooming. A central characteristic of the female's behavior during both of the pre-copulatory stages was that of hind leg vibration (*Loher, 1959*; *Strong & Amerasinghe, 1977*; *Uvarov, 1977*). During pre-mounting, leg vibration was mostly low-amplitude, with less frequent intermittent high- amplitude vibration. This is in accordance with Loher's contention (*1959*) that the amplitude of this element reflects the level of excitement of the locust (although, the role of this behavioral element in both sexes is still uncertain). It was also suggested that the female's vibration of her hind legs may serve as a defensive response against the male's mounting attempts. This is in accord with our major finding, suggesting that the most prominent behavioral elements demonstrated by the females were those related to rejection of the males.

## Sexual conflict

In this study we paid particular attention to the behavior of females and males at the points of possible conflict, preceding mate selection/decision. We suggest two points at which the conflict between the sexes is manifested (traffic lights in Fig. 3): the first occurs during pre-mounting and the second during the mounting stage. The first point of conflict may actually appear repeatedly before a male's attempts to mount the female, and is manifested in two elements: (1) the female's walking away ("running away" in *Loher, 1959*), and (2) jumping away (*Popov, 1958*; also referred to as "leaping away" in *Strong & Amerasinghe, 1977*). Jumping away better expresses rejection as it frequently followed dislodgment of the male. We did not include kicking during pre-mounting, although intuitively it may serve as a primary rejection element, because kicking is a common reflexive response of locusts, of both sexes, to tactile stimuli by other locusts, regardless of sex (*Norris, 1962*; *Siegler & Burrows, 1986*).

When attempting to mount the female, males displayed two behavioral elements: (1) climbing (described in this work for the first time), or (2) jumping (the more dominant
behavior, previously referred to as "attempt to copulate", "sexual attack", "copulation attack" or "assault"; (*Uvarov, 1928*; *Husain & Mathur, 1946* cited in *Popov, 1958*; *Loher, 1959*; *Pener, 1967a*; *Pener, 1967b*; *Otte, 1970*). These two elements were often preceded by peering or scanning (lateral swaying of the body from side to side). In both larvae and adult locusts this behavior is related to estimating distance (*Kennedy, 1945*; *Wallace, 1959*). Though not necessarily related to sexual interactions, scanning plays an important role in the pre-mounting stage, serving the males when jumping, and also in the females' rejection response to an approaching male.

Although, as mentioned above, the display of short stridulation was not very frequent during pre-mounting, its appearance was commonly associated with dislodgement of the male by the mounted female (in agreement with *Loher, 1959*). Based on their differential relative appearance during pre-mounting and mounting, our findings suggest different functional roles for the short and the long stridulation. The overall role of auditory signaling in the courtship behavior of the male desert locust, although previously considered as relatively insignificant (*Loher, 1959*; *Keuper et al., 1985*; *Robinson & Hall, 2002*) would thus appear to be worth revisiting.

In the second point of conflict, during the mounting stage, the interactions between the sexes were more complex. The females used both, direct and indirect rejection elements. Direct rejection comprised jumping and kicking (defensive reaction, *Loher, 1959*), commonly performed immediately after the male had mounted the female, and often presented sequentially, promoting dislodgement of the male from the female's back (repulsing the male, *Loher, 1959*). In response to the female's kicking behavior, a few males displayed defensive behavioral elements, including avoidance and blocking. These latter two elements, described here for the first time, may have a major role in assisting the male to overcome female rejection.

The indirect rejection by the female (passive phase, *Strong & Amerasinghe, 1977*), comprising her abdominal bending and abdominal grounding (the latter described here for the first time), is of special interest as it drew a distinctive response from the male: i.e., pressing his hind legs firmly to the ground in an attempt to lift the female.

We examined the efficacy of female jumping and kicking in successfully rejecting males at this conflict point by preventing the females from using their hind legs. Constraining the females indeed resulted in fewer mounting attempts and increased male mounting success. Hence we can safely postulate that a major component of mate-choice by the female is based on consistent and vigorous rejection by way of jumping and kicking. Males, however, overcome female rejection mostly by repeated mounting attempts.

Throughout this study we did not detect any clear signal of female receptivity. High receptivity was best demonstrated passively, whereby passive females did not reject the male (*Popov, 1958*). Twisting of the abdomen, suggested by *Ballard, Mistikawi & Zoheiry* (*1932*; cited in *Popov, 1958*) as a display of receptivity, was never observed in the current study. Another issue that has remained unresolved is that of inter-sexual recognition prior to pre-copulatory behavior. Previous reports have suggested mainly visual, but also chemical, signaling as playing a role in mutual recognition between the sexes in the desert locust (*Popov, 1958*; *Uvarov, 1977*; *Pener & Shalom, 1987*; *Obeng-Ofori, Torto & Hassanali, 1993*;

*Francke & Schmidt, 1994*; *Inayatullah, El Bashir & Hassanali, 1994*; *Ould Ely et al., 2006*). Our findings support a major role of visual signals, as we observed that rapid movement by the females (fast walking or jumping) appeared to enhance the males sexual stimulation.

## Concluding remarks

A detailed investigation of sexual and reproductive behavior is a prerequisite for understanding the evolutionary and ecological dynamics of a species (*Kirkendall, 1983*; *Thornhill & Alcock, 1983*). The comprehensive description presented here of the reciprocal interactions between the sexes in the desert locust thus contributes to our understanding of the biology and behavior of this economically significant pest. The described and presented ethogram offers a tool with which to compare behavioral similarities and differences among different orthopteran insects (*Paranjape, 1985*), and specifically among locust species. Here we exclusively described the sexual behavior of the desert locust in the gregarious phase. The knowledge acquired in this study and the tools developed for it will be used for a future comparative investigation of locusts in the gregarious and solitary phases, emphasizing the different features of the sexual conflict in relation to the phase phenomenon.

As noted, the desert locust is one of the most notorious agricultural pests. Major efforts have been invested in investigating the sexual behavior of pest insects (*Walgenbach & Burkholder, 1987*; *Rojas et al., 1990*; *Zahn et al., 2008*), with the rationale being that a better understanding of their sexual and reproductive behavior will contribute to the application of pest management (*Boake, Shelly & Kaneshiro, 1996*; *Suckling, 2000*). This work may thus also assist in identifying novel targets and generating environmentally friendly methods for locust control.

### Funding

This work was funded by a grant from the Israel Ministry of Agriculture and Rural Development (891-0277-13). The work in its final stage was partially supported by the German Research Council (DFG; Grant RI 2728/2-1). The funders had no role in study design, data collection and analysis, decision to publish, or preparation of the manuscript.

### Grant Disclosures

The following grant information was disclosed by the authors:
Israel Ministry of Agriculture and Rural Development: 891-0277-13.
German Research Council: RI 2728/2-1.

### Competing Interests

The authors declare there are no competing interests.

### Author Contributions

- Yiftach Golov performed the experiments, analyzed the data, prepared figures and/or tables, authored or reviewed drafts of the paper.

- Jan Rillich analyzed the data, prepared figures and/or tables, authored or reviewed drafts of the paper.
- Ally Harari and Amir Ayali conceived and designed the experiments, contributed reagents/materials/analysis tools, authored or reviewed drafts of the paper.

## Data Availability

All raw data extracted from behavioral experiments is provided in the Supplemental Files.

## Supplemental Information

Supplemental information for this article can be found online at http://dx.doi.org/10.7717/peerj.4356#supplemental-information.

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
