# Peer review of "Precopulatory behavior and sexual conflict in the desert locust"

_PeerJ, doi:10.7717/peerj.4356_

## Round 0.1 · original submission · Minor Revisions

· Academic Editor

Minor Revisions

Please revise your manuscript according to the reviewers' suggestions.

·

Basic reporting

Clear and unambiguous, professional English used throughout..

Experimental design

Original primary research within Aims and Scope of the journal.

Validity of the findings

Data is robust, statistically sound, & controlled. Conclusion are well stated, linked to original research question & limited to supporting results.

Additional comments

Although the Desert locust is serious pest, their sexual behaviors have not been extensively examined. The present study provided fundamental information of sexual behaviors by generating ethogram based on detailed observation and showed sexual conflict. Experiments were systematically conducted and all figures, tables and illustrations were clear. Especially, they have clearly demonstrated “mating partner choice” in females by modifying female behavior with simple experimental technique. They also revealed how males copulate with females on her back. This manuscript has been well written and appropriately covered most important references. I think this is the first report demonstrating sexual conflict in locusts based on qualitative and quantitative data. The authors categorized various behaviors, so it will help to unify terminology. Publication from open journal such as PeerJ is highly valuable for scientists. I believe this manuscript will be cited by many researchers. This manuscript has been already acceptable. I will give a few minor comments.

Comment 1
L225 insert “hind” before leg

Comment 2
L375 & Fig. 6
The authors concluded that female sexual behavior was not clear. I agreed with this conclusion, but I am interested in how negative female’s behaviors (walking away or jumping away) influenced on the rate of kicking behaviors when males tried to mount on her. The authors observed courtship behavior of a couple within a container for 3 h, so some males might try to mount on evading females which jumped or walked away from the male. According to my experience, some gregarious females tended to accept male’s mating trial without rejecting behaviors when gregarious females approached to males and physically contacted with males by their antennae (less than 20 %). I guess females which displayed negative behavior at S3 (walking away or jumping away) tended to reject male’s mating trial. It is interesting to compare the rate of rejection between females displayed negative S3 and positive S3.

Comment 3
Tables
The authors used “”pre-mounting”, but they used “pre mounting” (without “-”) in table

Comment 4
L89 & L441 Please insert “Ould” before Ely.


I am looking forward to seeing future publications comparing solitarious and gregarious locusts by using this publication.

Reviewer 2 ·

Basic reporting

I suggest to make some small improvements:
- To note somewhere taxonomic position of the species (family, subfamily) and its author(s)
line 42 - to add Orthoptera, Acrididae
line 60 analyses-the > analyses -- the
line 86 Acridids > acridids
line 131 my guess is that electrical power of lamps per se (25W) is not very important, data concerning illuminations are essential!
line 213 Figure 1 denote > Figure 1 denotes
line 218 antenna > antennae
line 234 an antagonistic nature > antagonistic nature
line 238 a Markov process > the Markov process
line 378 comprisedpalp > comprised palp
line 382 intermitetenthigh > intermittent high

Experimental design

no comments

Validity of the findings

no comments

---

## Round 0.2 · Minor Revisions

· Academic Editor

Minor Revisions

I have only a few minor suggestions, which I have entered on the manuscript (PDF version appended). Please approve those and resubmit your manuscript (PeerJ staff will send you the original Word doc).

---

## Round 0.3 · accepted · Accept

· Academic Editor

Accept

Thank you for the revisions. I am now recommending that your manuscript be accepted for publication- congratulations!